# Bone Marrow Stromal Cells-Induced Drug Resistance in Multiple Myeloma

**DOI:** 10.3390/ijms21020613

**Published:** 2020-01-17

**Authors:** Roberto Ria, Angelo Vacca

**Affiliations:** Section of Internal Medicine “G. Baccelli”, Department of Biomedical Sciences and Human Oncology, University of Bari “Aldo Moro” Medical School, Bari, 70124 Bari, Italy; angelo.vacca@uniba.it

**Keywords:** drug-resistance, microenvironment, multiple myeloma, plasma cells, stromal cells

## Abstract

Multiple myeloma is a B-cell lineage cancer in which neoplastic plasma cells expand in the bone marrow and pathophysiological interactions with components of microenvironment influence many biological aspects of the malignant phenotype, including apoptosis, survival, proliferation, and invasion. Despite the therapeutic progress achieved in the last two decades with the introduction of a more effective and safe new class of drugs (i.e., immunomodulators, proteasome inhibitors, monoclonal antibodies), there is improvement in patient survival, and multiple myeloma (MM) remains a non-curable disease. The bone marrow microenvironment is a complex structure composed of cells, extracellular matrix (ECM) proteins, and cytokines, in which tumor plasma cells home and expand. The role of the bone marrow (BM) microenvironment is fundamental during MM disease progression because modification induced by tumor plasma cells is crucial for composing a “permissive” environment that supports MM plasma cells proliferation, migration, survival, and drug resistance. The “activated phenotype” of the microenvironment of multiple myeloma is functional to plasma cell proliferation and spreading and to plasma cell drug resistance. Plasma cell drug resistance induced by bone marrow stromal cells is mediated by stress-managing pathways, autophagy, transcriptional rewiring, and non-coding RNAs dysregulation. These processes represent novel targets for the ever-increasing anti-MM therapeutic armamentarium.

## 1. Introduction

Despite the therapeutic progress achieved in the last two decades with the introduction of a more effective and safe new class of drugs (i.e., immunomodulators, proteasome inhibitors, monoclonal antibodies), without an improvement in patient survival, multiple myeloma (MM) remains a non-curable disease. [1,2,3,4,5,6]

Moreover, change in the therapeutic approach moving toward a long-term treatment, with the goal of providing continuous disease suppression, improves responses and survival without effect on disease curability. [7,8]

Relapsed patients remain not easy to treat, because the disease tends to become more aggressive, they develop drug resistance, and each relapse shortens their response duration [2,3,4,5].

MM is a B-cell lineage cancer in which neoplastic plasma cells expanding in the bone marrow (BM) and pathophysiological interactions with components of the microenvironment influence many fundamental biological aspects of the malignant phenotype (i.e., apoptosis, survival, proliferation, invasion) [9,10,11,12]. These interactions are mediated by autocrine and paracrine cytokines loops, and by cell–cell and cell–extracellular matrix (ECM) direct interactions [12,13,14,15,16]. Thus, regulating multiple signaling pathways plays one of the most important roles in the epigenetic control of the malignant phenotype and disease progression [9,10,17]. 

This review will be focused on the role of the BM microenvironment in the developed drug resistance of multiple myeloma during the course of the disease.

## 2. The BM Microenvironment

The BM microenvironment is a complex structure composed of cells, ECM proteins, and cytokines, in which tumor plasma cells home and expand [12]. The role of the BM microenvironment is fundamental during MM disease progression because its modification induced by tumor plasma cells is crucial for composing a “permissive” environment that supports MM plasma cells’ proliferation, migration, survival, and drug resistance [12]. In fact, all the biological processes active in the BM (i.e., angiogenesis, immune cell inhibition, osteoclasts activation, etc.) are functional to MM progression and drug resistance [18].

Moreover, BM stromal cells and non-cellular components (fibronectin, hypoxia, lactic acidosis, and nutrient withdrawal) promote protective endoplasmic reticulum (ER) stress-mediating drug resistance to melphalan and bortezomib [19].

### 2.1. The “Vascular Niche”

In the pathologic BM, endothelial cells collaborate with other cells to assemble a “vascular niche” (Figure 1) in which tumor plasma cells are protected from the aggression of anti-myeloma drugs and the immune system [20].

In the BM of MM patients with active disease, the endothelial cells display a typical phenotype characterized by the expression on their cellular surface of receptors (i.e., VEGFR-2, FGFR-3, cMET, and Tie2/Tek), increased expression of the β3-integrin, expression of endoglin, and expression of a water transporter, namely aquaporin 1 [21,22]. This “activated” phenotype is functional to the prevention of apoptosis, adhesion to the ECM, proliferation, migration, capillarogenesis, and enhanced interaction of plasma cells with the new-formed blood vessels, favoring plasma cells entry into circulation and later dissemination [20].

The expression of CD133 on a subset of BM endothelial cells during the active phase of the disease is indicative of the recruitment of CD133+ progenitor cells, derived from a common progenitor namely hemangioblast, which contributes to the neovascularization by means of the reactivation of the ancestral phenomenon named “vasculogenesis” [23,24,25,26,27].

Moreover, under the influence of MM microenvironmental and plasma cell factors, such as hypoxia, inflammation, expression of multiple cytokines, and growth factors, etc., MM endothelial cells switched to an angiogenic phenotype by means of down or upregulation of various crucial genes and related proteins [28,29].

Other cell types contribute to angiogenesis activation and are maintained during the active phase of MM. Hematopoietic stem cells that reside in the endosteum niche play a pivotal role in the regulation of vasculogenesis and angiogenesis, contributing to the protective niche in which plasma cells reside by alterations of the signals in the microenvironment [12,20,23,30].

Mesenchymal stem cells, of unclear origin in MM [31], represent an important component of the “vascular and osteoblast niche” [31,32,33]. They are potentially able to differentiate into multiple histotypes (i.e., fibroblasts, adipocytes, chondrocytes, and osteoblasts) and can migrate towards tumor sites modulating MM growth and playing a critical role in disease outcome [31,33]. There is evidence of mesenchymal stem cells-induction of bortezomib resistance in MM plasma cells by increasing Bcl2 expression and enhancing NF-κB activity via cell–cell contact [34,35]. 

Also, the BM adipocytes contribute to angiogenesis stimulation producing various angiogenic cytokines: VEGF, Ang-1 and -2, leptin, adiponectin, TNFα, FGF, tumor growth factor-β (TGFβ), HGF, interleukin (IL)-6, and IL-8 [36,37,38,39].

In patients with active MM, monocytes/macrophages under the signaling generated by plasma cell VEGF and FGF-2 production that bind VEGFR-1 and FGFR-1, -2, and -3 expressed on their surface, undergo a phenotypic and functional adaptation which contributes to building neovessels through vasculogenic mimicry [40,41].

Another cell typically involved in the inflammatory response, the mast cell can be recruited in the tumor bed where it can secrete several angiogenic factors contained in the granules: tryptase and chymase, heparin, histamine, TGF-β, TNF-α, IL-8, FGF-2, and VEGF [42,43,44]. Moreover, in the new vessels wall typical tryptase-positive mast cells connected by a junctional system with the endothelial cells can be evidenced (vasculogenesis mimicry) [45].

MM plasma cells reside in the vascular niche where they are protected by drug aggression [46]. The same cell composing the stroma surrounding plasma cells becomes resistant to anti-myeloma drugs in the advanced phase of the disease [47,48,49,50,51]. In endothelial cells, the nuclear stabilization of Hipoxia Inducible Factor (HIF)-1α induced by reactive oxygen species (ROS) is demonstrated in the BM of relapsed/refractory MM patients [51]. The nuclear stabilization of HIF-1α induces the expression of its target genes (i.e., VEGF, IL-8, and Osteoprotegerin (OPN)), inducing the proangiogenic phenotype maintenance as well as resistance to immunomodulators and proteasome inhibitors of endothelial cells [51]. The role of HIF-1α nuclear stabilization is confirmed by evidence that its inhibition impairs the MM plasma cells/stromal cells’ communication and angiogenesis-related functions as well as reverts bortezomib- and lenalidomide-resistance of endothelial cells [49,50,51]. This experimental evidence is followed by clinical evidence that the addition of the histone-deacetylase inhibitor, panobinostat, to bortezomib and dexamethasone increases the response in heavily pretreated, bortezomib-refractory MM patients [52,53]. The anti-myeloma activity of the histone deacetylase inhibitor panobinostat, particularly in the relapsed/refractory setting, is probably related to its indirect inhibition of the nuclear stabilization of HIF-1α [51]

### 2.2. The “Osteoblastic Niche”

Bone pain and hypercalcemia represent two of the major symptoms of MM and are a result of unbalanced bone remodeling caused by MM plasma that induces osteolytic lesions [17]. In the BM of MM patients, resident macrophages and plasma cells are induced to transdifferentiate into functional osteoclasts by release of IL-6, IL-1α or –1β, IL-11, TNF-α, TNF-β, and Macrophage-Colony Stimulating Factor (M-CSF) (also named osteoclast activating factors or OAF), as well as alterations of Runx2 and Wnt pathways [20,32,54]. Moreover, bone resorption is promoted by differentiation and activation signals in osteoclast precursors induced by the receptor activator of nuclear factor ligand (RANKL), the decoy receptor osteoprotegerin (OPG), its receptor (RANKR), and the chemokine macrophage inflammatory protein-1α (MIP-1α) [55]. In the osteoblastic niche, β1 integrins and Vascular Cell Adhesion Molecule (VCAM)-1 cadherins mediate the binding of MM plasma cells to stromal cells, overexpression of RANKL, and suppression of osteoprotegerin production favoring bone resorption [55]. Finally, there is evidence of CD38 expression on stromal cells and osteoblasts indicating the role of CD38 in osteoclast differentiation and bone resorption [56,57]. This evidence, with the observation that the ectoenzymatic network CD73/CD203a is active in the MM bone niche in the alternative production of adenosine (ADO) [58], suggests the possibility of a role of CD38 in the bone niche. In the osteoblastic niche (Figure 2), as in the vascular niche, the complex interactions with BM milieu influence the development of resistance to anti-myeloma therapy. It has been demonstrated that, by binding to a TNF-related apoptosis-inducing ligand (TRAIL), osteoprotegerin represents a paracrine survival factor by preventing drug-induced apoptosis in myeloma cells [59,60]. Moreover, stromal cells confer potent drug-resistance to blunt the efficacy of anti-myeloma drugs also by TGF-β-mediated inhibition of osteoblasts differentiation [61,62,63]. Indeed, terminally differentiated osteoblasts potentiate cytotoxic effects of melphalan and dexamethasone [64], suggesting that mature osteoblasts can overcome the drug resistance of MM plasma cells mediated by stromal cells into the osteoblastic niche.

### 2.3. Cancer-Associated Fibroblasts (CAFs)

In the stromal microenvironment of solid and hematologic cancer resides a subtype of fibroblasts with a typical phenotype, characterized by peculiar surface molecules and activated behavior, namely “cancer-associated fibroblasts” (CAFs) [65,66,67,68,69,70,71]. They derive from resident fibroblasts, from progenitor cells, and from cells undergoing the endothelial–mesenchymal transition (EndMT) or mesenchymal transition (MT) [70].

In the BM of MM patients, an important interplay between CAFs and plasma cells during MM initiation and progression has been demonstrated [71]. MM plasma cells induce and maintain the CAF-activated phenotype, which, in turn, supports myeloma progression by promoting pro-tumoral microenvironment modifications (i.e., extracellular matrix remodeling and angiogenesis) and by the direct induction of plasma cells’ proliferation and apoptosis resistance [71].

CAFs play a key role in the drug-resistance of MM cells. Frassanito et al. [72] demonstrated the role of autophagy in the cell–cell interaction between MM plasma cells and CAFs. Bortezomib induces the secretion of tumor growth factor-β (TGFβ). TGFβ is responsible for the conversion of normal fibroblasts into CAFs [73] and enhances the autophagic pathway. In refractory patients, CAFs show intrinsic activation of autophagy and their treatment with the autophagic inhibitor 3-Methyladenin (3-MA) or with TGFβ inhibitors increases apoptosis and restores their sensitivity to bortezomib. These observations suggest a pro-survival role of autophagy in CAFs and the ability of the same CAFs to protect PCs from bortezomib [72].

The production of fibroblast activation proteins (FAPs) by BM CAFs of MM has been also demonstrated [74]. FAPs are involved in tumorigenesis, progression, angiogenesis, and distance spreading of tumor cells as well as in drug resistance [75,76,77,78]. Through the activation of the β-catenin signaling pathway in MM cell lines, FAPs produced by BM stromal cells protect the same MM cells from bortezomib-induced apoptosis.

The communication between BM stromal cells and myeloma plasma cells is also mediated by microvesicles and exosomes that convey proteins and miRNA. Human BM stromal cells-derived exosomes increase cell viability and induce drug resistance to bortezomib of MM cells [79]. In fact, another way CAFs protect MM plasma cells from anti-myeloma drug is the dysregulation of miRNAs [80]. Accumulating evidence suggests that the dysregulation of miRNAs and exosomal miRNAs influences the crosstalk between cancer cells and the tumor microenvironment [81,82,83]. In BM fibroblasts of MM patients, aberrant miRNA profile plays a role in the progression of MM and in the change to a permissive microenvironment induced by myeloma plasma cells [84]. 

### 2.4. Adipocytes

Adipocytes residing in the BM directly interact with myeloma plasma cells [39] and protect them against apoptosis induced by anti-myeloma chemotherapy [85,86,87]. MM plasma cells induce BM resident adipocytes to release lipids, which are subsequently utilized as energy for tumor cell proliferation [39]. Moreover, adipocytes produce and secrete adipokines, that stimulate MM plasma cell growth through activating JAK/STAT-PI3K/AKT pathway [87], and resistin, which inhibits myeloma cell apoptosis and promotes ATP-binding cassette (ABC) transporters expression [86]. ABC overexpression is a well-known mechanism of multidrug resistance in various cancers [88]. Soluble adipokines produced by BM resident adipocytes activate autophagy in MM cell autophagy via JAK/STAT3 signaling [85]. As reported above, autophagy is a protective process through which MM plasma cells protect themselves from unfolded or misfolded proteins [89]. 

### 2.5. Immune Cells

In the BM microenvironment, the interaction between all cell components is influenced by the immune cells (T cells, NK cells, dendritic cells, myeloid-derived suppressor cells, etc.). These cells influence myeloma growth, survival, and progression (Figure 3).

In the MM pathologic BM, there is an imbalance of T lymphocyte subsets that play a relevant role in MM progression [90,91].

CD4+ T cells comprise several subsets: T helper 1 (Th1), Th2, Th17, and CD4+, CD25+, and T regulatory (Treg) cells [92]. All these T cell subsets secrete different cytokines with different functions during the immune response [93]. Th1 cells stimulate the cell-mediated immune response by interferon-gamma (IFN-γ) production [93], Th2 cells inhibit the Th1 cell-mediated response by IL-4 production [93], Th17 secretes IL-17A, IL-6, and TNF-α that play a role in inflammation response [94] and have been implicated in MM occurrence and progression and its complications [95], and Treg cells produce and secrete TGF-β and IL-10 that inhibit effector T cell growth and exert immunomodulatory actions [94]. In the MM BM, there is a higher infiltration of CD4+, Th1, and Th17 subsets indicating a switch towards immunotolerance [96].

Another T cell type, Th22 cells, is significantly represented in the BM of MM patients, particularly in the advanced stage of the disease [97]. They are implicated in adaptive immune responses and are identified by the production of IL-22 and IL-13, but not IL-17, and are significantly increased in the BM of stage III and relapsed/refractory MM [98].

The switch toward a self-tolerance of immune responses against tumor cells in the BM of MM patients is also suggested by the Treg activity that participates in the MM-related immune dysfunction [99]. Treg cells were reported as dysfunctional and modified in their proportion between peripheral blood and BM in myeloma patients particularly in relapsed/refractory patients [99,100].

A small subset of T cells, the γδ T cells, is one protagonist of both innate and adaptive features [101]. Despite that γδ T cells generally reside and work in tissues, no differences have been found in the peripheral blood (PB) or the BM of MM patients [102]. γδ T cells show the ability to simultaneously act directly through their anti-myeloma cytotoxic activity and indirectly by stimulating or regulating the biological functions of other cell types, such as dendritic cells and cytotoxic CD8+ T cells [101,103]. 

The cytotoxic lymphocytes natural killer cells (NK), involved in the innate immune response and anti-tumor immunity, as well as natural killer T (NKT) cells, sharing properties of both T cells and NK cells show impaired differentiation and function in the BM of MM patients [90,104,105]. Parallel to MM progression the loss of ligand-dependent IFN-γ production in NK and NKT cells which disables the cytotoxicity capacity of host resistance, has been demonstrated [104].

Dendritic cells (DC) process and present antigens to T cells and serve as a critical link between the innate and adaptive arms of the immune system. Two distinct subsets of DCs have been defined, namely, the myeloid DC (CD11c+) and the plasmacytoid DC (CD11c- CD123+) [106]. It has been demonstrated that DC function and DC distribution are both impaired in patients with myeloma and this impairment is related to drug resistance of plasma cells [107,108].

Myeloid-derived suppressor cells (MDSCs) are a heterogeneous population of cells showing suppressor activity on T cell response that expands during cancer, inflammation, and infection [109]. The increase of CD14^+^/HLA-DR^−/low^ MDSCs in newly diagnosed MM patients and during the active/advanced phase of the disease have been reported [110,111]. Arginase-1 (Arg-1) is the principal mediator of the suppressive function of MDSCs [112]. Arg-1 depletes the extracellular matrix of the essential amino acid, arginine [112], and directly promotes MM progression and plasma cell resistance to therapy [111]. The inducible isoform of NOS (iNOS) is also induced by MDSCs [112]. iNOS blocks T cell activity through nitration of tyrosine residues inhibiting their phosphorylation and thus T cell function [113].

## 3. Therapeutic Targeting of the Microenvironment and Clinical Application

The multiple interactions between BM components (cellular and non-cellular) of the microenvironment and MM plasma cells seem an ideal approach for the treatment of MM patients [46]. There are numerous pieces of evidence on the dual anti-myeloma activity, both on MM plasma cells and on the microenvironment, of currently used drugs. The proteasome inhibitors (i.e., bortezomib, carfilzomib, ixazomib) targeting the proteasome complex act on key cellular processes, such as cell cycle progression, inflammation, immune surveillance, growth arrest, and apoptosis, by means of the modulation of NF-κB transcription factor [114]. NF-κB also regulates the expression of adhesion molecules, such as ICAM-1 and VCAM-1, on both MM cells and BM stromal cells [115], and controls the production of IL-6 by BM stromal cells that increases production and secretion of VEGF-2 and FGF-2 from MM plasma cells [116]. Blocking NF-kB proteasome inhibitors inhibits MM cell adherence to the BM stromal cells, reducing MM cell growth and VEGF-2 and FGF-2 secretion [11,12,15,16,20]. Moreover, proteasome inhibitors downregulate the production and secretion of angiogenic cytokines (VEGF, IL-6, IGF-I, Ang-1, and Ang-2), targeting neovessel formation through the inhibition of activated endothelial cells [117]. Finally, proteasome inhibitors act on osteoblastic niches stabilizing β-catenin in pre-osteoblasts enhancing osteoblastogenesis and negatively regulating RANKL [118].

Immunomodulators (IMIDs such as thalidomide, lenalidomide, and pomalidomide) near their direct tumoricidal activity, have antiangiogenic properties [119], modulate TNF-α signaling [46,119], reduce angiogenic cytokines [46,51,71], interfere with NF-κB activity [119], and disrupt the direct interactions between MM plasma cells and BM stromal cells by modulation of cell surface adhesion molecules [120]. Moreover, lenalidomide can overcome resistance in patients with relapsed or refractory MM [121].

As previously reported, MM plasma cells are protected in the vascular niche and in advanced phases of the disease, the same stromal cells acquire resistance to anti-myeloma drugs as demonstrated by the nuclear stabilization of HIF-1α in the BM endothelial cells of relapsed/refractory MM patients [51]. This resistance can be reverted by the addition of HDAC inhibitors (i.e., vorinostat or panobinostat) to other anti-myeloma drugs [52,53].

There are pieces of evidence that the use of anti-resorptive drugs (bisphosphonates) in patients with MM inhibits angiogenesis [122] and is associated with benefits in terms of skeletal-related event rates as well as in terms of the progression-free survival rate of myeloma patients [123].

The most successful therapeutic approach to MM in the last years is represented by the introduction of immunotherapy (IMIDs, monoclonal antibodies) for the treatment of myeloma patients. This approach aims to contrast the immune-permissive microenvironment contrasting the inhibitory signals and enhancing the cytotoxic activity of effector cells [90,91,92,93,94,95,96,97,98,99,100,101,102,103,104,105,106,107,108,109,110,111,112,113]. IMIDs increase the number and function of immune cells, particularly NK cells, NKT cells, and T cells, increase Th1 cytokines production (e.g., IFN-γ and IL-2) that increase the cytotoxic activity of the immune system, activate T and natural killer cells which may lead to apoptosis of myeloma cells by antibody-mediated cellular cytotoxicity (ADCC), and enhance immune surveillance [124]. Recently, two monoclonal antibodies were approved for the treatment of MM patients, anti-CD38 and anti-SLAMF7 [125]. Both of these antibodies act by mean induction of ADCC, antibody-dependent cellular phagocytosis (ADCP), and complement-dependent cytotoxicity (CDC), and show immunomodulatory effects (block regulatory cells and myeloid-derived suppressor cells) [125]. Moreover, the anti-CD38 antibody daratumumab possesses direct cytotoxicity activity on MM plasma cells mediated by blocking the CD38 signaling pathway [126].

The new promising horizon of immunotherapy is the targeting of immune inhibitory molecules and pathways, such as programmed cell death 1 and its ligand (PD-1/PD-L1), lymphocyte activation gene 3 (Lag-3), B-cell maturation antigen (BCMA), cytotoxic T lymphocyte antigen-4 (CTLA-4), and T cell immunoglobulin, and ITIM domain (TIGIT) [127,128]. Clinical studies of PD-1/pd-L1 inhibitors in relapsed myeloma and combination therapy with other anti-myeloma drugs are ongoing. Finally, some studies with adoptive T cell therapy with genetically modified T cells with chimeric antigen receptors, namely CARs or CAR-T cells [129], for the treatment of MM patients are ongoing with promising results [130].

## 4. Conclusions

MM is a disease with multiple faces that is extremely complex. The recurrent drug-resistant phenotype is parallel to disease progression and can arise quickly or later during its natural history. Stromal cells of the BM microenvironment play a pivotal role in the progression, disease evolution, and drug-resistance onset. Numerous modifications involve the BM stromal cells, which contribute to the beginning and sustenance of the plasma cells drug-resistant phenotype. The identification of pivotal players in the BM environment can lay the groundwork for future treatments that are less toxic and more effective. The final goal should target malignant plasma cells and stromal support in BM niches with the aim to eradicate the disease and avoid the onset of drug-resistance.

## Figures and Tables

**Figure 1 ijms-21-00613-f001:**
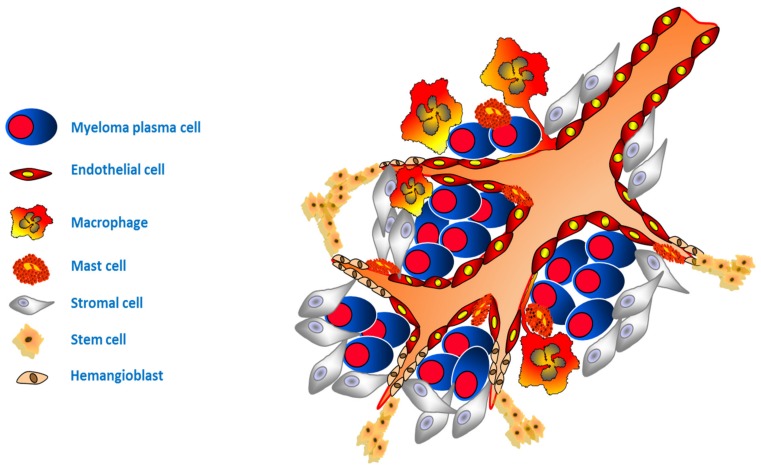
The “vascular niche”. In the pathologic bone marrow (BM), endothelial cells collaborate with other subtypes of stromal cells to assemble the vascular niche in which multiple myeloma (MM) plasma cells are stimulated to proliferate and survive, and are protected from the aggression of anti-myeloma drugs and immune system.

**Figure 2 ijms-21-00613-f002:**
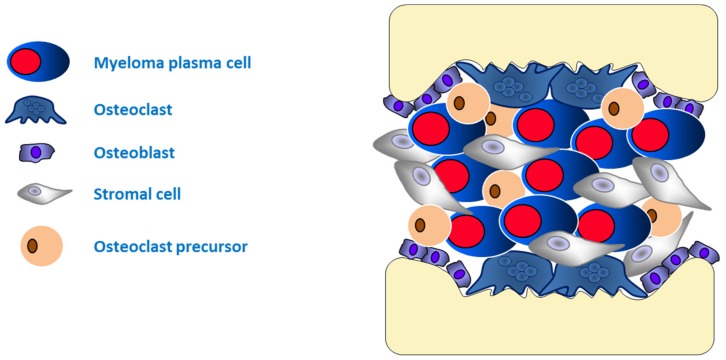
The “osteoblastic niche”. In the pathologic BM, resident macrophages and plasma cells induce the activation of osteoclasts and osteoclast precursors differentiation to assemble, with osteoblasts and the other subtypes of stromal cells, the osteoblastic niche in which MM plasma cells are stimulated to proliferate and survive, and are protected from the aggression of anti-myeloma drugs and the immune system.

**Figure 3 ijms-21-00613-f003:**
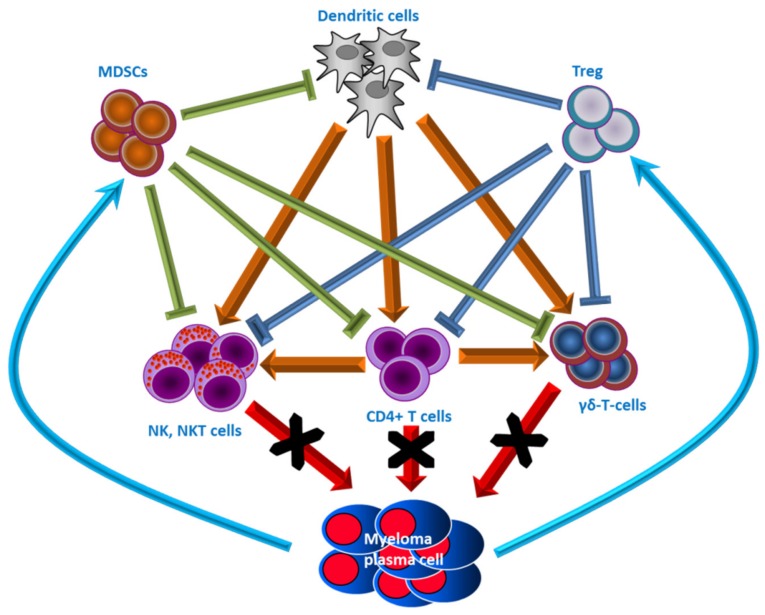
Schematic of functional interactions of immune cells and with malignant plasma cells. The anti-myeloma activity of T cells, natural killer cells (NK), natural killer T cells (NKT), and γδ T cells against MM plasma cells is inhibited by malignant plasma cells via the activation of T regulatory cells (Tregs) and stimulation of myeloid-derived suppressor cells (MDSCs). Moreover, dendritic cell activity is also inhibited by Tregs and MDSCs. This makes the microenvironment immunologically permissive towards myelomatous proliferation and induces MM plasma cell escape from immune surveillance.

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
