# Peer review of "Bone Marrow Stromal Cells-Induced Drug Resistance in Multiple Myeloma"

_ijms, 2020, doi:10.3390/ijms21020613_

Round 1

Reviewer 1 Report

In this review, the authors summerized "Bone marrow stromal cells-induced drug resistance in multiple myeloma". Multiple myeloma is a B-cell lineage cancer in which neoplastic plasma cells expanding in the bone marrow were the pathophysiological interactions with the components of microenvironment influence many biological aspects of the malignant phenotype, apoptosis, survival, proliferation, invasion. This review summerized the important part of this field.

The authors should provide the theraputic targeting of this field. Is any potential treatment of this topic ? The authors should be provided. The clinical application should be discussed.

Author Response

According to reviewer criticism, a new paragraph: “Therapeutic targeting of the microenvironment and clinical application” has been added in the revised text.

Reviewer 2 Report

The manuscript describes in a comprehensive manner the drug resistance in multiple myeloma induced by bone marrow stromal cells. The authors have revised the relevant literature on the topic and organized information in a logical order. The manuscript is well written and provides all readers with important information, presented in a highly readable manner. I was able to identify however some minor errors in the use of the English language, such as:

Row 39: "Relapsed patients remains not easy to treat, because of the..." should be "Relapsed patients remain not easy to treat, because of the"

Row 139-140: "Have been demonstrated that, by binding to TNF-related apoptosis-inducing ligand, osteoprotegerin represent a paracrine survival factor by preventing drug-induced apoptosis in" - the sentence cannot begin with a verb

Row 185: "Adipocytes resident in the BM directly interacts with myeloma plasma cells" should be "Adipocytes resident in the BM directly interact with myeloma plasma cells"

and so on.

Therefore I recommend the publication of the current paper after a complete revision of the English grammar and spelling.

Author Response

(The authors gave the same response as above.)

Round 2

Reviewer 1 Report

Accept to publish 

This manuscript is a resubmission of an earlier submission. The following is a list of the peer review reports and author responses from that submission.